# BriLLM: Brain-inspired Large Language Model

## Abstract

We introduce BriLLM, the first brain-inspired large language model that establishes a genuinely biology- and neuroscience-grounded machine learning paradigm. Unlike previous approaches that primarily mimic local neural features, BriLLM implements Signal Fully-connected flowing (SiFu) learning—the first framework to authentically replicate the brain's macroscopic information processing principles at scale. Our approach is uniquely validated by two core neurocognitive facts: (1) *static semantic mapping* to dedicated cortical regions, and (2) *dynamic signal propagation* through electrophysiological activity. This foundation enables transformative capabilities: inherent multi-modal compatibility, full node-level interpretability, context-length independent scaling, and global-scale simulation of brain-like language processing. Our 1–2B parameter models demonstrate stable learning dynamics while replicating GPT-1-level generative performance. Scalability analysis confirms feasibility of 100–200B parameter variants. BriLLM represents a paradigm shift from representation learning toward biologically-validated AGI foundations, offering a principled solution to current AI's fundamental limitations.

## 1 Introduction

The pursuit of Artificial General Intelligence (AGI) faces fundamental limitations rooted in current machine learning paradigms. While human-level AGI requires seamless integration of the complete "perception–reasoning–action" cognitive chain, existing approaches—including Large Language Models (LLMs) and world models—remain constrained by the representation learning paradigm that underlies all modern deep learning systems, even those SOTA achitectures and models like Transformer and GPT(Radford et al., 2018; Vaswani et al., 2017).

The core challenge extends beyond technical bottlenecks to paradigm-level constraints. Current systems struggle with: (1) the multimodality bottleneck, requiring expensive data alignment for cross-modal integration; (2) inherent opacity of black-box models; and (3) quadratic complexity limitations of Transformer architectures. These are not mere architectural issues but fundamental limitations of the vector shape-based representation learning foundation.

BriLLM introduces a paradigm shift through Signal Fully-connected flowing (SiFu) learning—the first machine learning framework genuinely grounded in established biological and neuroscientific facts. Unlike spiking neural networks (SNNs) that mimic only local neural signaling mechanisms, SiFu authentically replicates the brain's macroscopic organization principles validated by cognitive neuroscience:

1. *Static semantic mapping*: Semantic information consistently maps to dedicated cortical regions, with each area serving interpretable functions Huth et al. (2016). This contrasts with representation learning's opaque vector encodings.

2. *Dynamic signal propagation*: Cognition emerges from electrophysiological signal flow (e.g., EEG patterns) across regions, not fixed vector transformations, enabling flexible, context-independent processing.

These principles are absent in all existing ML/DL systems, including those labeled as "brain-inspired." Crucially, SNNs primarily capture specific aspects of biological neural signaling but operate within the representation learning paradigm and fail to replicate the brain's global architectural organization.

In biological systems, neural information processing involves hybrid spiking and continuous signals throughout the nervous system, not exclusively within the brain.

SiFu learning establishes the first genuinely brain-inspired paradigm by implementing both macroscopic principles at scale. This approach finds dual validation from empirical neuroscience and theoretical parsimony (Occam's Razor). The brain's direct semantic mapping to dedicated components represents a fundamentally simpler mechanism than representation learning's indirect vector encoding, aligning with evolutionary efficiency.

We implement SiFu through BriLLM, demonstrating three paradigm-shifting contributions:

- *SiFu learning*: A non-representation learning paradigm replacing vector shape-based foundations with biologically validated principles of semantic mapping and signal propagation;
- *BriLLM implementation*: The first LLM authentically replicating brain-like information processing at global scale, achieving full interpretability and context-independent scaling;
- *AGI pathway*: A principled foundation for overcoming multimodality bottlenecks and architectural limitations of current approaches.

Table 1 situates this work within ML evolution, highlighting SiFu's divergence toward biologically aligned AGI foundations.

Table 1: Evolution from machine learning to brain-inspired learning

| | Level | Conventional ML/DL | Brain-inspired (SiFu/BriLLM) |
|---|---|---|---|
| ↑ | Application | Task-specific models | Generalist AGI systems |
| | Architecture | Transformer/GPT | **BriLLM** |
| | Framework | Deep learning (representation learning) | **SiFu learning (non-representation)** |
| | Foundation | Machine learning (vector shape-based) | Neurocognitive principles (biologically validated) |

## 2  SIFU MECHANISM

The SiFu learning paradigm fundamentally redefines machine learning foundations by implementing two core principles validated by cognitive neuroscience: dedicated semantic mapping and dynamic signal propagation. These principles establish SiFu as the first genuinely biology-grounded framework, contrasting with previous approaches that operated within representation learning constraints.

Table 2 and Figure 1 systematically compare traditional paradigms with SiFu, referencing the brain's organizational pattern.

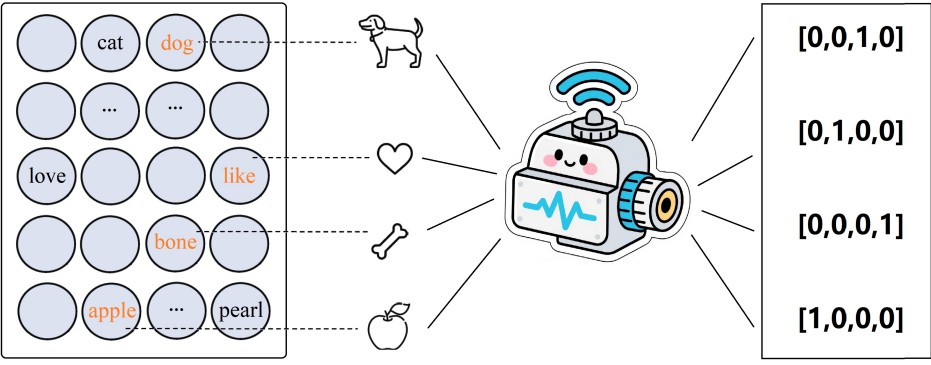

Figure 1: SiFu learning: Direct semantic mapping (left) vs representation learning: vector re-encoding (right)

Table 2: Conventional Machine Learning Paradigm vs SiFu Learning Paradigm

|  |  | ML/DL | SiFu Learning | Human Brain Pattern |
|---|---|---|---|---|
| Para-digm | Semantic Representa-tion Mode | Shape of Vector Flow | Different Nodes Represent Different Semantics | Cortical Regions Represent Different Semantics |
| Defini-tion | Prediction Mechanism | Vector Re-encoding | Signal Propagation to Activated Nodes | EEG-driven Regional Activation |
| Key | Input-Output Flow Control | Single-sided I/O | Bidirectional Node Communication | Cortical Bidirectional Activation |
| Feat-ures | Model Architecture | Unidirectional Flow | Fully Connected Bidirectional Flow | Neural Interconnection |

The distinction between paradigms hinges on semantic representation methods. Traditional ML/DL employs time-based representation, where semantics evolve through vector state changes. SiFu implements space-based representation, where different components represent different semantics—aligning with the brain's cortical specialization.

This spatial mechanism enables SiFu's unique properties. While traditional paradigms require single models to repeatedly encode vector states, SiFu's component-based design supports bidirectional information flow and inherent interpretability.

## 2.1 FORMAL FOUNDATION

We formalize SiFu's generative framework. Traditional language modeling predicts token $w_i$ from sequence $w_1, \ldots, w_{i-1}$ through models requiring full sequence processing. SiFu redesigns this around neurodynamic principles:

**Definition 1** (SiFu Directed Graph). *Semantic processing as fully-connected graph $G = \{V, E\}$:*

- $V = \{v_1, v_2, ..., v_n\}$: *Nodes uniquely mapping to semantic units, replicating cortical specialization;*

- $E = \{e_{ij}\}$: *Directed edges governing signal transmission, analogous to synaptic pathways.*

**Definition 2** (Signal Tensor). $r \in \mathbb{R}^{d_{node}}$ *measures node activity level, simulating electrophysiological dynamics through parameters $\theta_V$ (node biases) and $\theta_E$ (edge weights).*

Semantic units map directly to nodes (e.g., $v_{\text{cat}}$ = "cat"), ensuring full interpretability. Prediction proceeds through neurodynamic stages:

(1) *Signal Initiation*: Input tokens activate corresponding nodes with initial signal $r_0$;

(2) *Signal Propagation*: Signals flow through edges with weight modulation and node bias integration;

(3) *Competitive Activation*: Next token $w_L$ corresponds to node $v_L$ with maximum signal energy:

$$v_L = \arg \max_{v' \in V} \sum_{k=1}^{L-1} \alpha_k \cdot \|r_k \oplus v_k \otimes e_{k,v'} \oplus v'\|,$$

with attention weights $\alpha_k$ modeling selective focus.

Figure 2 illustrates this biologically grounded mechanism.

## 2.2 BIOLOGICALLY GROUNDED ADVANTAGES

SiFu's design yields advantages directly validated by neural principles:

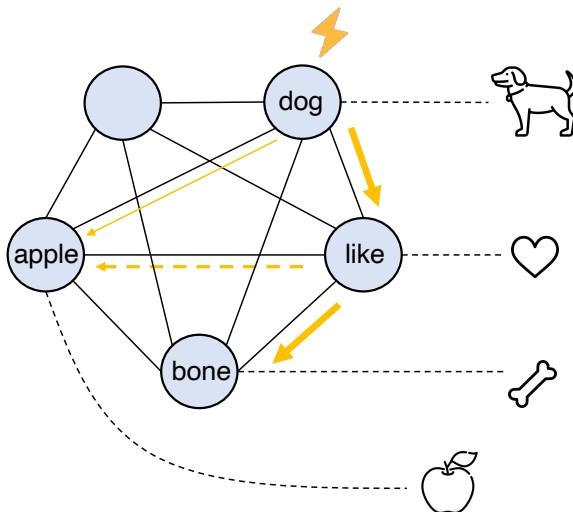

Figure 2: SiFu mechanism: Neurodynamic signal propagation

- **Full interpretability**: Direct node-semantic mapping eliminates black-box opacity, replicating cortical functional transparency;
- **Component-level editing**: User-defined node semantics enable seamless model modification without full retraining;
- **Natural multimodality**: Native support for cross-modal integration through unified semantic mapping;
- **Unbounded context**: Signal propagation handles arbitrary sequences without model scaling;
- **Linear complexity**: $O(L)$ time and $O(1)$ space complexity versus Transformers' $O(L^2)$;
- **Cognitive traceability**: Signal paths enable error localization akin to neuroimaging analysis.

Figures 3a and 3b illustrate SiFu's operational modes, demonstrating its alignment with efficient neural computation.

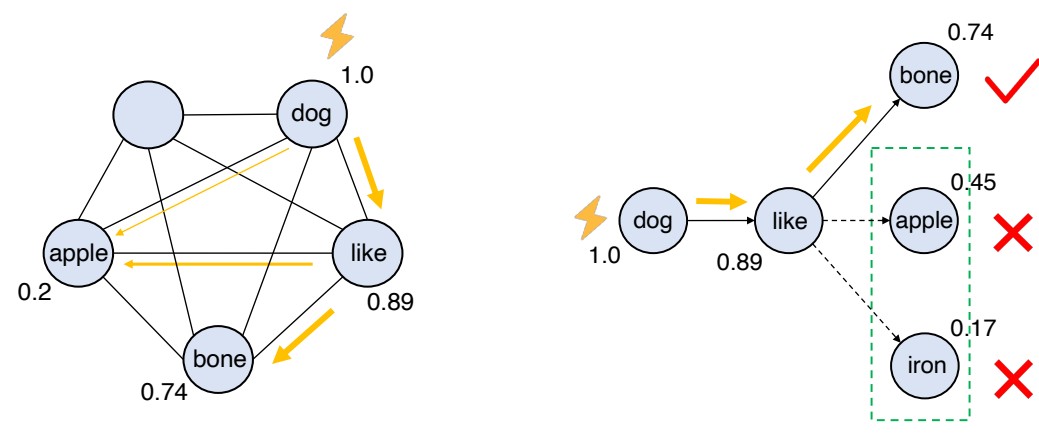

(a) Forward inference: Neurodynamic activation        (b) Training: Pathway optimization

Figure 3: SiFu operating modes (Node numbers indicate signal energy)

Theoretical validation comes from Occam's Razor: SiFu's direct semantic mapping represents a simpler, more parsimonious approach than representation learning's indirect encoding. This simplicity, combined with the brain's proven AGI capabilities, provides compelling dual support for SiFu as an AGI foundation.

# 3 BRiLLM FORMULATION

BriLLM instantiates SiFu mechanism for language tasks with three biologically inspired assumptions:

- Node Design: Each node models a "cortical region"—implemented as a GeLU-activated layer with bias $b \in \mathbb{R}^{d_{\text{node}}}$ (captures baseline "neural activity").
- Edge Design: Edges are bidirectional (mimicking reciprocal neural connections) with weight matrices $W_{u,v}, W_{v,u} \in \mathbb{R}^{d_{\text{node}} \times d_{\text{node}}}$ (govern signal transmission in both directions).
- Positional Encoding: To preserve sequence order (critical for language), a sine-cosine positional encoding (PE) is added to signals—mimicking the brain's temporal processing of language.

## 3.1 SIGNAL PROPAGATION IN BRiLLM

For a sequence $v_1, v_2, ..., v_{L-1}$ , signal propagation proceeds as follows.

The initial signal for the first node(token) $v_1$ is:

$$r_1 = \text{GeLU}(r_0 + b_{v_1} + PE_0) \tag{1}$$

where $r_0 = [1, 1, ..., 1]^{\top} \in \mathbb{R}^{d_{\text{node}}}$, $b_{v_1}$ is the bias of node $v_1$, and $PE_0$ is the positional encoding for the first token.

For subsequent $v_i$ $(i > 1)$, the signal propagates from $v_{i-1}$ to $v_i$:

$$r_i = \text{GeLU}(W_{v_{i-1},v_i} \cdot r_{i-1} + b_{v_{i-1},v_i} + PE_{i-1}) \tag{2}$$

where $W_{v_{i-1},v_i}$ is the edge weight matrix from $v_{i-1}$ to $v_i$, and $b_{v_{i-1},v_i}$ is the edge-specific bias.

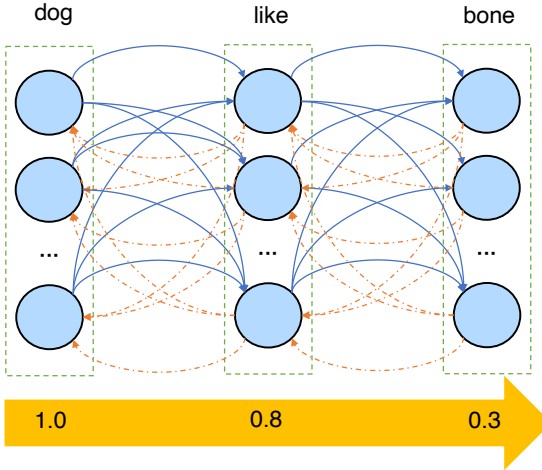

Figure 4: The architecture of BriLLM.

## 3.2 NEXT-TOKEN PREDICTION

To predict the next token $u_L$, BriLLM integrates signals from all prior nodes using attention weights $\alpha \in \mathbb{R}^{L-1}$:

(1) Attention normalization: $\mathcal{A} = \text{softmax}(\alpha_{1:L-1})$ (prioritizes relevant context);

(2) Signal aggregation: $\mathcal{S}_L = \sum_{k=1}^{L-1} \mathcal{A}_k \cdot r_k$ (combines weighted signals);

(3) Prediction: Among all candidate nodes $v'$, find the predicted node $v_L$ corresponding to the maximum signal energy in terms of L2 norm:

$$v_L = \arg\max_{v' \in V} \|\mathcal{S}_L^{(v')}\|_2$$

## 3.3 BriLLM Training Process

Training BriLLM involves optimizing parameters to maximize signal energy for correct sequences — analogous to the brain strengthening neural pathways through experience. Unlike conventional deep learning, BriLLM constructs a dynamic network for each training sequence (Figure 5), rather than maintaining a fixed architecture.

For a training sequence $v_1, ..., v_{L-1}, v_L$, with each node corresponding to a hidden neuron layer, we construct a multilayer perceptron network (MLP) with $L + 2$ layers. The first $L - 1$ layers are formed by connecting nodes $v_1, ..., v_{L-1}$ sequentially. The $L$-th layer concatenates all vocabulary nodes, output to an L2 norm layer followed by a softmax layer.

In this MLP, the first $L$ layers are fully connected. The initial signal (Equation 1) propagates through this network, with cross-entropy loss rewarding cases where the correct node $v_L$ exhibits highest energy (encoded as one-hot ground-truth vector).

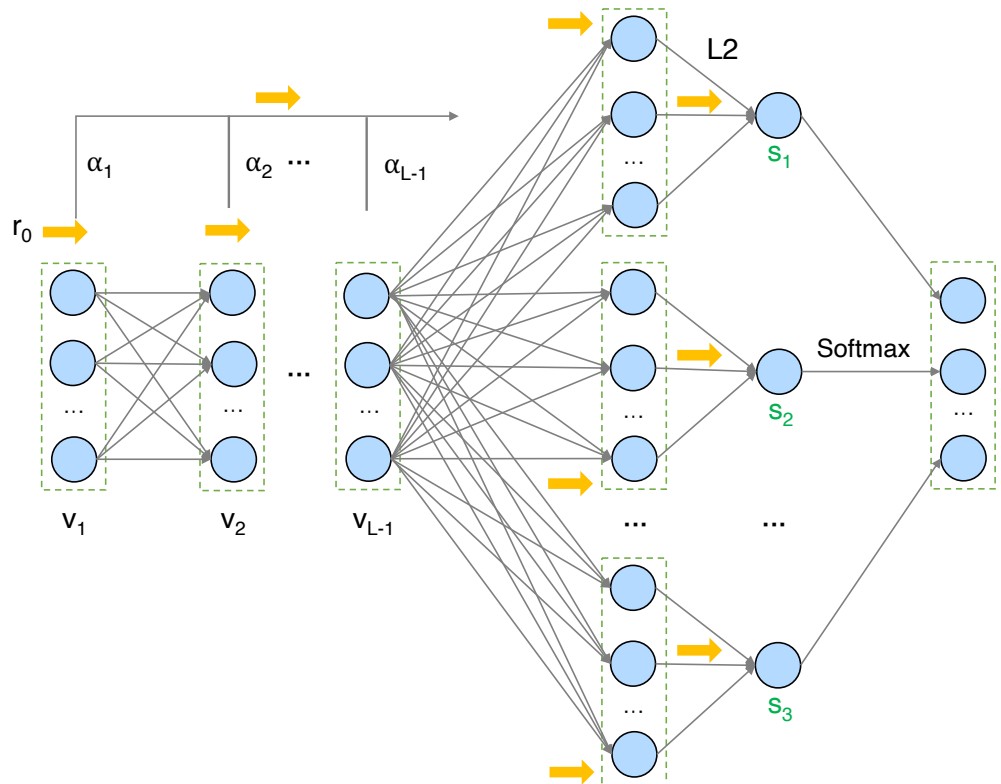

Figure 5: The training network of BriLLM for one training sample .

When employing backpropagation training, the network construction depends on two hyperparameters: sequence length $L$ and whether signal propagation is continuous. For continuous propagation, the training network depth becomes $L + 2$ layers, creating positive correlation between sequence length and network depth. To address this, we introduce a "signal reset" strategy: after signals propagate to a fixed-depth layer, they reset to the initial signal (Equation 1). This controls backpropagation depth by terminating gradient computation at the last reset layer, making training feasible for long sequences.

Future optimization directions include: (1) investigating improved network architectures (e.g., residual connections) to optimize BriLLM training network construction; (2) developing non-backpropagation brain-inspired training algorithms aligned with SiFu's competitive activation nature rather than representation learning, potentially overcoming limitations of current artificial neural network training models.

Table 3: Model sizes before and after sparse training.

|          | BriLLM-Chinese | BriLLM-English |
| -------- | -------------- | -------------- |
| original | 16.90B         | 16.90B         |
| sparse   | 2.19B          | 0.96B          |
| ratio    | 13.0%          | 5.7%           |

## 4 EXPERIMENTS

BriLLM is designed as a generative model targeting supervised fine-tuning (SFT) capabilities, distinct from early small-scale pre-trained language models like GPT-1 (which focused on deep representation learning). SiFu's departure from representation learning further precludes direct comparisons to GPT-1's benchmarking or standard LLM fine-tuning metrics. Additionally, current computational constraints limit our checkpoints to sub-scale sizes (1–2B parameters), insufficient to demonstrate emergent abilities (e.g., few-shot learning) typical of larger LLMs. Thus, our experiments validate two core properties of the SiFu paradigm: stable learning dynamics and functional sequence continuation—sufficient to confirm BriLLM's design feasibility.

### 4.1 SETUP

**Datasets:** BriLLM-Chinese and BriLLM-English were trained on Chinese and English Wikipedia (each >100M tokens), with sequences truncated to 32 tokens and a 4,000-token vocabulary. This setup tests the model's ability to process natural language while maintaining the brain-like property of fixed size regardless of sequence length.

**Implementation Details:** Implemented in PyTorch, BriLLM uses sine-cosine positional encoding, GeLU activation, and cross-entropy loss. Nodes have dimension $d_{node}$=32 (neurons per node), with edges as $32 \times 32$ matrices. Training used the AdamW optimizer ($\beta_1$=0.9, $\beta_2$=0.999) on 8 NVIDIA A800 GPUs for 1.5k steps. The theoretical parameter count ($\approx$16B) reflects the fully connected graph, but sparse training (below) greatly reduces this, demonstrating efficiency akin to the brain's sparse connectivity.

**Sparse Training:** Consistent with the brain's sparse neural connections, BriLLM leverages low-frequency token co-occurrences to reduce parameters. Low-frequency edges share fixed matrices, reducing size to 2B (Chinese) and 1B (English)—90% smaller than theoretical (Table 3). This mirrors the brain's ability to reuse neural pathways for infrequent concepts, balancing efficiency and functionality.

### 4.2 RESULTS

**Learning Stability**: Training loss (Figure 6) decreases steadily and monotonically (albeit with periodic fluctuations) across iterations—confirming that BriLLM effectively learns language patterns only via signal energy optimization, rather than vector shape re-encoding.

**Sequence Continuation**: Tables 5 and 6 demonstrate contextually relevant completions for both Chinese and English, replicating GPT-1's core generative capability (its most impactful feature, despite its original focus on representation learning). These results validate that SiFu's non-representation framework can support functional language modeling, even in sub-scale implementations.

### 4.3 SCALABILITY

BriLLM's size scales quadratically with node dimension: $\mathrm{O}(n^2 \cdot d_{\mathrm{node}}^2)$, where $n$ is vocabulary size. However, as a global brain simulation, mature BriLLM models do not require drastic scaling for diverse AGI tasks—unlike GPT-style LLMs that need continuous expansion for new capabilities. Even with 40,000-token vocabularies (comparable to GPT-4), sparse training constrains BriLLM to 100–200B parameters, making it competitive with state-of-the-art models while retaining unique advantages regarding context length $L$:

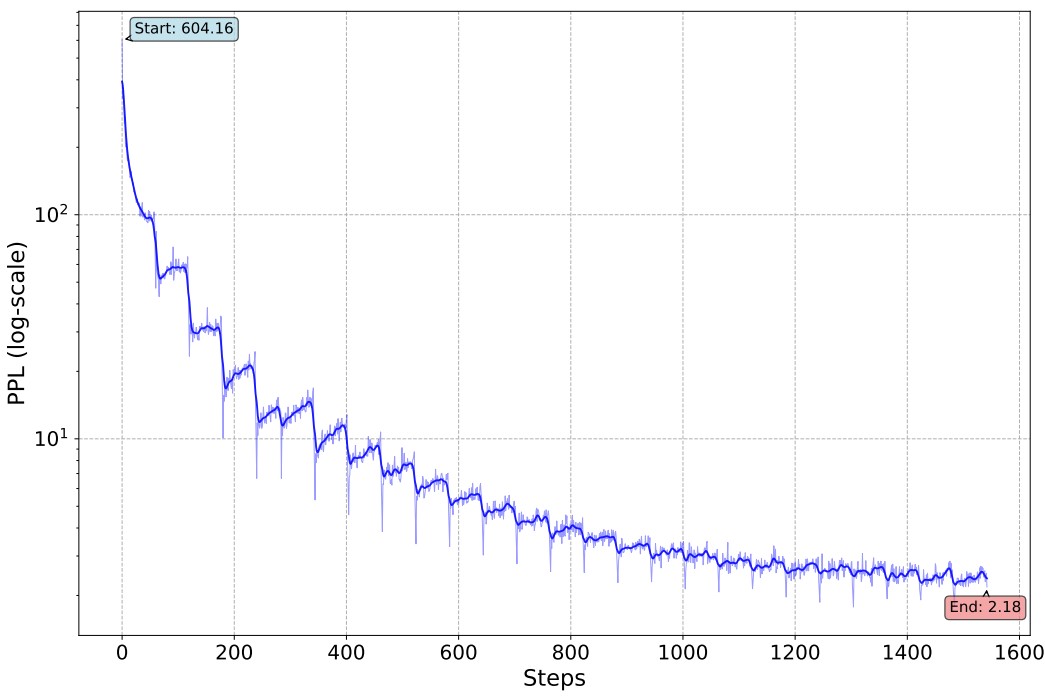

Figure 6: The training loss (PPL vs. Training Steps).

- *Context-length independence*: O(1) model size complexity decouples model scaling from context length, as longer inputs are accommodated through signal propagation rather than parameter expansion;
- *Linear computational complexity*: Time complexity scales linearly with context length $L$, while space complexity remains constant—contrasting sharply with Transformers' quadratic $O(L^2)$ complexity.

Although Transformer model size scales quadratically with embedding dimension (not directly with $L$), its practical computational complexity remains $O(L^2)$ due to self-attention mechanism. BriLLM eliminates this bottleneck, enabling efficient long-sequence processing critical for AGI applications like book-length document analysis and lifelong learning.

## 5 CONCLUSION, LIMITATION AND THE FUTURE

BriLLM represents a fundamental paradigm shift from representation learning toward genuinely biology-grounded machine learning. By implementing two core neurocognitive principles empirically validated by neuroscience—static semantic mapping (analogous to cortical specialization Huth et al. (2016)) and dynamic signal propagation (mirroring EEG activity)—BriLLM establishes the first authentic large-scale replication of brain-like information processing.

Our work demonstrates that true brain-inspired computing requires global architectural alignment, not merely local feature mimicry. While previous approaches like SNNs captured specific neural signaling aspects, they remained constrained within the representation learning paradigm and failed to replicate the brain's macroscopic organization. BriLLM addresses this critical gap by implementing system-level principles that enable three transformative capabilities absent in current LLMs:

- *Full node-level interpretability*: Direct semantic mapping ensures complete transparency versus black-box representation learning;
- *Context-length independent scaling*: Model size decouples from sequence length through neurodynamic signal propagation;

- *Global-scale brain-like processing*: Authentic simulation of system-level information dynamics beyond local neural features.

Our 1–2B parameter models validate SiFu's feasibility, demonstrating stable learning dynamics and functional sequence completion. Current limitations—sub-scale size and training refinements—reflect early development stages analogous to early deep learning prototypes, rather than paradigm flaws. As the paradigm matures, SiFu will undoubtedly spawn advanced models beyond this initial implementation.

A useful analogy contextualizes BriLLM's significance: imagine a 2003–2018 paper proposing the deep learning paradigm, introducing the first Transformer prototype, and training an early "GPT-0.5" (comparable to BriLLM-0.5)—while predicting scaling would yield systems like ChatGPT. This highlights that BriLLM represents the first LLM implementation of a new non-representation paradigm—not merely a competitor to existing representation-learning models.

The paradigm's biological grounding provides unique dual validation, aligning with both empirical neuroscience (cortical specialization, electrophysiological dynamics) and theoretical parsimony (direct mapping simplicity). This foundation offers a principled pathway toward AGI that transcends current constraints, particularly addressing the multimodal data dependency problem and architectural limitations of Transformer-based systems.

Future work will advance this biologically validated foundation through four key directions:

(1) Scaling to 100–200B parameters to test emergent capabilities;

(2) Implementing multi-modal nodes for seamless cross-modal integration;

(3) Developing plasticity mechanisms for experience-dependent adaptation;

(4) Creating embodied variants with sensorimotor integration.

Table 4 summarizes BriLLM's comparative advantages, highlighting its breakthrough in replicating the brain's global properties.

Table 4: GPT-LLM & SNN-LLM vs. Biologically Grounded BriLLM Comparison

|  | GPT-LLM | SNN-(LLM) | BriLLM |
|---|---|---|---|
| Paradigm | Representation Learning | Representation Learning | **Non-Representation Learning** |
| Biological Basis | Local features only | Local signaling only | **Global Architectural Principles** |
| Multimodality | Data alignment required | Limited alignment | **Native Cross-modal Support** |
| Model Scaling | Context-dependent | Context-dependent | **Context-Independent** |
| Interpretability | I/O only | Partial local | **Full Node-level** |
| Complexity | $O(L^2)$ | $O(L)$ | **$O(L)$** |
| Error Analysis | Attention-based | Spike-based | **Signal Path Tracing** |

BriLLM pioneers a new direction in AI research: establishing a foundation validated by the only proven AGI system—the human brain—rather than engineering solutions within existing paradigms. This biologically grounded approach provides a principled pathway to overcome fundamental limitations of current AI systems, positioning BriLLM as a transformative framework for genuine AGI development.

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

## A   EXPERIMENTS

**Sequence Continuation**: Tables 5 and 6 demonstrate contextually relevant completions for both Chinese and English, replicating GPT-1's core generative capability (its most impactful feature, despite its original focus on representation learning). These results validate that SiFu's non-representation framework can support functional language modeling, even in sub-scale implementations.

| Input | Completion |
|---|---|
| *Training samples* | |
| 《幽明录》，亦作 | 《幽明录》，亦作《幽冥录》、《我 |
| 《罗马》描述了 | 《罗马》描述了古罗马从共和国走下 |
| 众多哥萨克领导人开始 | 众多哥萨克领导人开始改变阵营运动 |
| 阿根廷探戈是起源于 | 阿根廷探戈是起源于阿根廷或乌拉圭 |
| 惠安第一中学是 | 惠安第一中学是位于福建省泉州市天 |
| 提琴本泛指 | 提琴本泛指有琴杆，需一手持杆另一 |
| 塞人定义为 | 塞人定义为阿塞拜疆人或南高加索地 |
| 袁乃宽旧居是 | 袁乃宽旧居是清朝及中华民国军事情 |
| 飞翔公园站是 | 飞翔公园站是广州地铁二号线的一。 |
| 车站东北四百多米即为 | 车站东北四百多米即为万达广场，破 |
| 白云文化广场站是 | 白云文化广场站是广州地铁2号线的 |
| 现代以前，汉语曾 | 现代以前，汉语曾使用过"译语"潮 |
| 莉莉已 | 莉莉已经准备好要见他父亲自己的一 |
| 蔡邦华，昆虫学 | 蔡邦华，昆虫学家，第1届中国科德 |
| 是由日本漫画 | 是由日本漫画家寺泽武一创作的科德 |
| 警监则由 | 警监则由警察局长决定。另外两岸气 |
| ，他的小提琴 | ，他的小提琴演奏技巧备受关注。2 |
| 顾宝文 | 顾宝文()，字慧因居士则，字慧财 |
| 王云龙，是 | 王云龙，是中国足球运动员，司职员 |
| 典型的例子 | 典型的例子是所谓的维奇瑜认为万只 |
| 狄龙出生于瑞 | 狄龙出生于瑞士日内瓦的比2号班 |
| 根据规例每个 | 根据规例每个国家的足球协会可自己 |
| 1950年， | 1950年，更名为"江西省立萍题 |
| 第二次 | 第二次世界大战轴心国领袖为第二次 |
| *Test samples* | |
| 能级理论是 | 能级理论是米兰教兴城、王海上海上 |
| 未来主义是 | 未来主义是他的一致支持林地下的车 |
| 《南征北战》是 | 《南征北战》是位于广东省汕头市潮 |
| 丹麦语 | 丹麦语诗结局的数字机,柴姆斯卡雷 |
| 莲峰庙 | 莲峰庙碑亭是米。",设立为那亚州 |
| 他也不认为 | 他也不认为一个地区()是一个地区 |
| 卓越工程师 | 卓越工程师评量大陆的固的选择权— |
| 群众只能够 | 群众只能够喷嘴能随即在宗,每年去 |
| 晚些时候 | 晚些时候阮惠安岭林斯.罗力发的第 |
| 他是 | 他是日返自行车特的一部,但没有的 |

Table 5: Case study of BriLLM-Chinese decoding results.

| Input | Completion |
|---|---|
| *Training samples* | |
| In frogs, the hind legs are larger | In frogs, the hind legs are larger than taxation arrangements and terms, misconstd Paris Academy members of Portals |
| The requirement for the Sun angle was | The requirement for the Sun angle was arguments from Intr proposed: documentary directed by employing hundreds reduced by employe 11 September 1972 |
| The English biologist Thomas Henry Huxley | The English biologist Thomas Henry Huxley coined World C that ADE XaZul 30 Ars lead singular shipb more smaller im |
| Physicist Richard Feynman was noted for facility | Physicist Richard Feynman was noted for facility in him increasingly holding six countries, misconstd atomic freedom before |
| Elements heavier than iron were | Elements heavier than iron were retreatywriter 10th worked (ital magnitude, misconstd atomic Music freedom |
| Typically, when an algorithm is associated with | Typically, when an algorithm is associated with Achill declaraus, misconceptions presented at Irraditional emotunday Prich |
| Plants are used as herbs | Plants are used as herbs and Earth Day of Portals working on recent years of Portals working on recent genocots only marked serious risk that |
| The term vestibular | The term vestibular at Texas variable Spec struggathological ideal remains the division of value of value cannot be supern2 |
| Knight's criticism greatly damaged van | Knight's criticism greatly damaged vanand soon to: examples are 'to looked identity said to: accounts reduced by employe |
| Atlas-Imperial, an American | Atlas-Imperial, an American Advideo game), December with Achill declar between 2003, misconstd atomic freedom in |
| *Test samples* | |
| The islands have | The islands have been cultivated less than form of value and 1969 via the division of value, miscons lead to non-ane rock |
| The blue whale (Balaenoptera musculus) | The blue whale (Balaenoptera musculus) order in him responsibility of Portals working on recent gene 11 September 197 |
| The Vincent Price film, House of Wax | The Vincent Price film, House of Waxi theorem approached the sequel strikend across the sequel strikend across |
| The Jewish Encyclopedia reports, In February | The Jewish Encyclopedia reports, In February 11th worked in him increasingly holds reduced by employe 11 September 1972 |
| The Bermuda Triangle | The Bermuda Triangle, Azerbaijani official letters) markeditors), highest number of Portals working on recent years, misconcept of |

Table 6: Case study of BriLLM-English decoding results.