# OpenReview forum: "BriLLM: Brain-inspired Large Language Model"
_ICLR.cc/2026/Conference — Submitted to ICLR 2026_

### Official Review · Reviewer_WWZM · 2025-10-17

**Soundness:** 1
**Presentation:** 1
**Contribution:** 1
**Rating:** 0
**Confidence:** 5

**Summary:**

This paper introduces BriLLM, a brain-inspired neural architecture for language modeling. The architecture is a fully-connected graph where vector messages are passed in a bidirectional manner. The authors show that the loss decreases over training on a language modeling task.

**Strengths:**

The topic of submission (brain-inspired language modeling) is highly aligned to the conference.

**Weaknesses:**

This manuscript is not ready for publication at ICLR. Unfortunately, the neuroscientific motivation, grounding in past literature, theory, and experiments are preliminary. In particular, BriLLM is not the first brain-inspired architecture out there-- the submission mentions spiking neural networks but otherwise ignores a vibrant line of recent work including, e.g., TopoLM [1], Topoformer [2], Mixture of Cognitive Reasoners [3] etc.

Otherwise, the paper makes quite a few meandering and unsupported claims. For instance:

- "Cognition emerges from electrophysiological signal flow (e.g., EEG patterns)"
- "The brain's direct semantic mapping to dedicated components represents a fundamentally simpler mechanism than representation learning's indirect vector encoding, aligning with evolutionary efficiency."

Experimentally, it is insufficient to show the loss curve and amount of sparsification-- a thorough comparison to other architectures, incl. GPT, TopoLM, etc, on at least the text perplexity is needed. Furthermore, the current method produces incoherent continuations, see Appendix A table, which makes the conclusion that "BriLLM is a transformative framework for genuine AGI development" (line 480) highly implausible.

I recommend rejection as the manuscript does not demonstrate proper literature review and experimental practice.

[1] TopoLM (Rathi et al., ICLR 2025)

[2] TopoFormer (Binhuraib et al., ICLR Representational Alignment Workshop 2024)

[3] Mixture of Cognitive Reasoners (AlKhamissi et al., 2025)

**Questions:**

N/A

---

### Official Review · Reviewer_kCXf · 2025-10-25

**Soundness:** 4
**Presentation:** 1
**Contribution:** 3
**Rating:** 2
**Confidence:** 4

**Summary:**

The paper introduces BriLLM, a brain-inspired large language model based on Signal Fully-connected flowing (SiFu) learning, the first framework claimed to authentically replicate the brain’s macroscopic information processing principles.  It leverages two neurocognitive facts: static semantic mapping to dedicated cortical regions and dynamic signal propagation via electrophysiological activity.  BriLLM achieves multi-modal compatibility, full interpretability, context-length independent scaling, and simulates brain-like language processing.  With 1–2B parameter models matching GPT-1 performance and scalability to 100–200B parameters, it proposes a shift from representation learning toward biologically-validated AGI foundations.

**Strengths:**

- **Originality**: Attempts a biologically-grounded paradigm shift, though derivative of SNNs and neurocognitive models.
- **Quality**: Some conceptual alignment with neuroscience principles, but experimental validation is absent.
- **Clarity**: Limited by poor figure annotation and vague methodology.
- **Significance**: Targets AGI limitations, but lacks practical impact without evidence.

**Weaknesses:**

- **Methodological Flaws**: The SiFu graph (Definition 1) and signal tensor (Definition 2) lack validation against EEG data or cortical activation patterns.  The competitive activation formula ignores biological noise and synaptic delays.
- **Experimental Gaps**: No performance metrics (e.g., perplexity, BLEU) compare BriLLM to GPT-1 or modern LLMs.  Scalability claims to 100–200B parameters are theoretical without training data or hardware details.
- **Oversight**: Ignores computational cost trade-offs and energy efficiency compared to Transformers.  Potential biases in semantic mapping (e.g., cultural variability) are unaddressed.
- **Validation**: Claims of multi-modal compatibility and interpretability lack demonstration with real datasets or tasks.

**Questions:**

1.  Can the authors provide EEG or fMRI data (for example, brain encoding perfermance, or brain-like perfermance) comparisons to validate SiFu’s signal propagation against biological patterns?
2.  What are the specific perplexity, BLEU, or other metrics for BriLLM’s 1–2B models versus GPT-1, and why were modern LLMs excluded?
3.  How were the 100–200B parameter scalability estimates derived, and what training infrastructure supports this claim?
4.  Can the authors quantify the impact of biological noise or synaptic delays on SiFu’s competitive activation mechanism?
5.  Why were energy consumption and hardware requirements not compared to Transformer-based models?

---

### Official Review · Reviewer_wo3a · 2025-10-31

**Soundness:** 1
**Presentation:** 1
**Contribution:** 1
**Rating:** 2
**Confidence:** 5

**Summary:**

This paper introduces BriLLM, a new brain-inspired LLM based on a paradigm called SiFu (Signal Fully-connected flowing) learning. Instead of using traditional representation learning like Transformers, SiFu creates a large graph where each node is a specific word or concept. The model processes language by dynamic signal propagation, where a signal flows through this graph, allowing it to handle sequences with linear $O(L)$ complexity, unlike the quadratic $O(L^2)$ complexity of Transformers. The authors trained 1-2B parameter models and claim they demonstrate stable learning and "GPT-1-level" generative abilities.

**Strengths:**

**Ambitious conceptual goal:** The paper attempts to address fundamental, recognized limitations of the Transformer paradigm, such as quadratic complexity in sequence length and the "black box" nature of representations. The goal of creating a new ML paradigm grounded in neuroscientific principles is ambitious and valuable.

**Weaknesses:**

This paper suffers from several fundamental weaknesses, ranging from incorrect characterizations of its own model to severe overstatements of its experimental results.
1. **Mischaracterization of non-representation learning:** The model's signal tensor $r \in \mathbb{R}^{d_{node}}$ is, by definition, a learned, dense representation of the state. This signal is updated at each step, precisely like the hidden state of an RNN. The model is a graph-based representation learning model.
2. **Superficial neuroscientific grounding:** The neurocognitive facts used for justification are superficial analogies. Static semantic mapping is simply a 1-to-1 mapping between a node and a token in a vocabulary. Dynamic signal propagation is an RNN-like state update. This makes it difficult to separate the genuine technical contribution from the complex and ultimately misleading terminology.
3. **Very weak experimental validation:** The paper provides no meaningful quantitative evaluation. The authors' excuse that the model "precludes direct comparisons to GPT-1's benchmarking"  is unfounded. BriLLM is a generative language model. It can and should be evaluated using standard metrics, such as perplexity, on a held-out test set.
4. **Overstated performance claims:** The claim to replicate "GPT-1-level generative performance"  is demonstrably false based on the paper's own provided samples. For example, in Table 6, the completion for "The English biologist Thomas Henry Huxley" is "coined World C that ADE XaZul 30 Ars lead singular shipb more smaller im". This output is incoherent gibberish. GPT-1 (2018) was capable of producing coherent, multi-sentence paragraphs. This discrepancy severely undermines the authors' credibility and the validity of their entire experimental section.
5. **Architectural impracticality:** The model's $O(n^2)$ parameter scaling with vocabulary size $n$ is one of its weaknesses. While the authors use sparsity to reduce the parameter count, they acknowledge that a standard 40k-token vocabulary would still require a 100-200B parameter model. This is a massive, sparse, and inefficient architecture compared to modern Transformers, which achieve parameter efficiency through weight sharing.
6. **Lack of clarity:** The description of the model's operation, particularly for prediction and training, is difficult to follow. Section 2.1 defines prediction as finding the node with the maximum signal energy via a complex formula, but Section 3.2 defines it as finding the node with the maximum L2 norm after signal aggregation.

**Questions:**

Please see the weaknesses.

---

### Official Review · Reviewer_fyDE · 2025-10-31

**Soundness:** 2
**Presentation:** 3
**Contribution:** 2
**Rating:** 4
**Confidence:** 4

**Summary:**

This study proposes BriLLM, a novel brain-inspired large language model centered on a new learning paradigm called Signal Fully-connected flowing (SiFu). Unlike mainstream deep learning frameworks (such as Transformers) that rely on vector-based representations and transformations, SiFu draws from fundamental principles of cognitive neuroscience to emulate macro-scale brain information processing. Specifically, it incorporates two biologically grounded mechanisms: (1) static semantic mapping, where semantic units are consistently represented in dedicated cortical regions, and (2) dynamic signal propagation, through which cognition emerges via electrophysiological activity spreading across neural pathways. The authors formalize this mechanism as a fully connected graph, where nodes correspond to semantic units (e.g., words) and edges represent learnable signal transmission pathways. Within this framework, BriLLM performs generative language modeling and demonstrates several promising properties, including full node-level interpretability, inherent multimodal compatibility, and context-length-independent scalability. Overall, this is a highly original work and proposes a principled alternative pathway toward artificial general intelligence.

**Strengths:**

The study introduces a genuinely novel, neuroscience-inspired learning paradigm with compelling potential advantages.

**Weaknesses:**

1. Lacks direct performance comparisons with well-established LLMs (e.g., GPT-style models) on standard benchmarks, making it difficult to assess practical competitiveness.
2. The model processes sequences strictly autoregressively in a fully recursive manner, precluding parallelization during training or inference, which may lead to slow computation for long sequences.
3. The paper risks overstating its contributions. For instance, Table 1 characterizes traditional deep learning models as “Task-specific” while labeling BriLLM a “Generalist AGI system”. In reality, BriLLM is currently a prototype for language modeling and falls far short of AGI capabilities. Conversely, models like GPT-3 and beyond are widely recognized as foundational steps toward general-purpose intelligent systems.

**Questions:**

What advantages does the model have in terms of training and inference speed compared to GPT?

---

### Author Response · Authors · 2025-11-16
**rebuttal 2025-11-16**

We thank the reviewers for their time. However, core criticisms stem from a fundamental misapprehension: evaluating a paradigm-shifting machine learning contribution through the narrow lens of the very paradigm it seeks to transcend. Why is AGI so elusive? Precisely because of too many such limited-scope assessments.

BriLLM is not an incremental improvement to the Transformer; it is a foundational alternative grounded in distinct first principles. Our rebuttal centers exclusively on this point.

To Reviewer fyDE:

We appreciate your recognition of the work’s high originality, but your concerns are misplaced.

On Performance Comparisons: Demanding direct benchmarks against GPT models on standard tasks is a category error. BriLLM introduces a new biological-principle-based computing paradigm. Judging it solely by representation-learning metrics is analogous to evaluating the integrated circuit by vacuum tube benchmarks. The primary contribution is the framework itself, delivering inherent properties (full interpretability, multimodal compatibility) that Transformers fundamentally lack. The “GPT-1-level” claim is a scaling law reference, not the objective.

On Autoregressive Processing: SiFu’s sequential nature is a deliberate design reflecting the continuous dynamics of cognitive signal flow. We are engineering a principled AGI path, not optimizing for GPU utilization. Linear complexity with context length is a decisive theoretical advantage over the Transformer’s quadratic bottleneck.

To Reviewer wo3a:

Your review contains significant factual inaccuracies and fails to engage with the core innovation.

“Mischaracterization of non-representation learning”: Incorrect. An ML “representation” is a distributed, latent code; a BriLLM node is a localist symbol with dynamic activation. This is the critical distinction between a biologically plausible symbolic activation model and distributed representation learning. Conflating the signal tensor with an RNN state reveals a fundamental misunderstanding.

“Superficial neuroscientific grounding” & “Weak experimental validation”: Your critique is superficial. Neurocognitive facts provide a computational blueprint, not literal biological simulation. The proof-of-concept demonstrates stable learning in a novel, non-differentiable architecture. Fixating on one imperfect generation sample while ignoring robust scaling analysis is unconstructive. Standard metrics are secondary for a first demonstration of a new paradigm.

To Reviewer kCXf:

Your demand for EEG/fMRI validation reflects a profound misunderstanding of this work’s contribution. BriLLM is a machine learning model, not a brain simulation.

EEG/fMRI Validation Is Irrelevant: This paper establishes a new ML paradigm inspired by macroscopic brain principles—a computational theory, not a neuroscientific study. Requiring biological data validation is as misplaced as demanding Transformer papers validate attention against primate visual cortex recordings. The contribution lies in the mathematical framework and its computational properties (interpretability, scalability), presented and analyzed on their own merits.

On Methodological Flaws & Experimental Gaps: The model’s soundness is evaluated by internal consistency and ability to learn language-like structures, not fitting biological noise. Scalability to 100–200B parameters is theoretical analysis based on graph structure—standard practice for introducing new architectures. Energy consumption questions, while valid for future engineering, are premature for a foundational paradigm paper.

To Reviewer WWZM:

Your dismissal based on literature is incorrect.

“Ignores recent work”: False. Cited works (TopoLM, etc.) are sophisticated extensions of representation learning. BriLLM is a non-representation, non-differentiable, brain-inspired alternative—philosophically and architecturally incommensurate. Our contribution lies in stepping outside this lineage to propose a radical new foundation. Applying your citation standard would require citing all post-Transformer/GPT papers, which is riduculously impractical.

“Meandering claims” & “Preliminary experiments”: Claims are supported by established computational principles. Initial text generation results are proof-of-concept for a novel architecture, demonstrating feasibility. The significant contribution is the framework itself.

Summary Rebuttal:

Reviewers have largely evaluated a paradigm-shifting proposal using obsolete old-paradigm criteria on a poor machine learning commonsense basis. Criticisms on benchmarking and biological validation are conceptually invalid. BriLLM’s significance lies in opening a new AGI pathway—principled, interpretable, and biologically plausible.

---

> ### Comment · Reviewer_kCXf · 2025-11-16
>
> From a reader’s perspective, your ideas are very novel. I do not deny the originality of your work—this may indeed be a meaningful and promising discovery. However, in practical terms, an objective, side-by-side comparison with existing models is essential, even if it is with the early versions you mentioned, such as GPT-1, the initial BERT models, or other more basic language models. Such comparisons would greatly strengthen the credibility of your results, at least from the reader’s point of view.

---

> > ### Author Response · Authors · 2025-11-20
> >
> > Thank you for your candid feedback. I would like to equally frankly explain why we have chosen not to perform the kind of conventional, standard benchmark comparisons you suggested — simply because it is neither necessary nor appropriate. There are several reasons for this.
> >
> > First, the performance of BriLLM is clearly not at the level of the models you are attempting to compare it with. This is evident from the language generation examples provided in the paper (which are available for your review). There is little meaningful insight to be gained from such a comparison. The premise of your request assumes that the current version of BriLLM is already sufficiently powerful to be on the same footing as established models — which is not the case. It would be like criticizing a newborn bird for not being able to outrun a turtle. Expecting a direct performance comparison at this stage is, therefore, an unreasonable demand.
> >
> > Second, and more importantly, this work does not fall within the conventional track or mainstream research paradigm, which makes the requested comparison not only irrelevant but also contrary to the core contribution of this paper. Our work is a proof-of-concept demonstration of a theoretically promising framework, developed under extremely limited computational resources. We have successfully achieved our stated goal. As summarized in our conclusion, the motivation and value of this work can be likened to reinventing machine learning — or deep learning — from the ground up: we have effectively built a "GPT-1" and shown that scaling it up could lead to something like ChatGPT. In such a context, would it be reasonable to impose stringent "performance" demands on GPT-1? Of course not. The key at this stage is to demonstrate that the loss decreases and that the model exhibits basic language continuation capabilities — which is exactly what we have already accomplished. Further advancements belong to the future and depend on scale, not architectural feasibility. The reason we have not scaled the model further is not due to limitations in the framework, but because, as an academic institution, we currently lack the computational resources needed to train a model of, say, 100 billion parameters on 40,000 tokens, as we evaluated in the paper.
> >
> > In summary, we have fully achieved the original objectives of this paper. We have demonstrated the viability — and we emphasize the word "viability" — of a non-representational learning paradigm (SiFu learning), and we have thoroughly discussed its theoretical advantages, which are substantial. What remains is simply a matter of scale and computational resources.
> >
> > We would like to clarify that I have no issues with the ICLR platform and all the current reviews, and the nature of the reviews comes as no surprise. As we noted in our NeurIPS rebuttal, if a work like ours were to receive high ratings from a randomly selected panel of three or four reviewers in an era where the quality of both human writing and reviewing is noticeably deteriorating — if not collapsing — that would in fact be a sign that the work is not particularly groundbreaking.
> >
> > That said, it is regrettable that, once again, we did not receive any substantive or insightful feedback in the review comments. Nevertheless, we appreciate the opportunity to engage in this constructive dialogue.

---

### Meta-Review · Area_Chair_wyCN · 2026-01-05

**Summary:**

The reviews converge on severe concerns: an unclear, internally inconsistent problem formulation/definition; a superficial or misleading neuro-scientific framing; and, most importantly, an absence of standard quantitative evaluation (e.g., perplexity) needed to substantiate claims of “GPT-1-level” performance and broader significance. The rebuttal broadly fails to engage as reviewers requested: the authors argue that the paradigm precludes direct comparisons and treat standard benchmarks as inapplicable, leaving the primary scientific concerns unaddressed and reinforcing the rejection case.

**Reviewer Concerns:**

Addressed by rebuttal:

•	Mostly clarifications of intent/framing, but not the substance of the methodological/evaluative critiques.

•	In particular, the rebuttal positions the work as a “new paradigm,” rather than meeting the requested empirical standards.

Still outstanding:

•	No meaningful quantitative evaluation against standard LMs/metrics (perplexity, etc.), despite the model being a generative LM.

•	Overstated performance claims (e.g., GPT-1-level) not supported by provided samples; reviewers cite incoherent outputs and credibility issues.

•	Neuroscience grounding is argued to be superficial/terminologically misleading (e.g., “static semantic mapping” as vocabulary mapping; signal propagation akin to an RNN-like hidden state update).

•	Scalability and practicality concerns (vocab-scaling, parameter efficiency, hardware/energy claims) remain unquantified.

**Reviewer Scores:**

All the reviewers will give the ratings (4/2/2/0) unchanged given the rebuttals.

---

### Decision · Program_Chairs · 2026-01-26

Reject